# Next Steps in Prader-Willi Syndrome Research: On the Relationship between Genotype and Phenotype

**DOI:** 10.3390/ijms232012089

**Published:** 2022-10-11

**Authors:** Joyce Whittington, Anthony Holland

**Affiliations:** Department of Psychiatry, University of Cambridge, Douglas House, 18b Trumpington Road, Cambridge CB2 8AH, UK

**Keywords:** Prader-Willi syndrome, genotype, phenotype, hypothalamus, brain development

## Abstract

This article reviews what we know of the phenotype and genotype of Prader-Willi syndrome and hypothesizes two possible paths from phenotype to genotype. It then suggests research that may strengthen the case for one or other of these hypotheses.

## 1. Introduction

In the short-term, research in Prader-Willi syndrome (PWS) aimed at alleviating symptoms is more likely to achieve quick results and provide some benefit to people with the syndrome and their families. However, in the long term, research needs to concentrate on elucidating the mechanisms whereby the absence of the expression of specific gene(s) gives rise to the particular symptoms associated with PWS so that interventions can be more precisely targeted. Our aim in this article is not to stifle the search for treatments to alleviate symptoms, indeed in [1] we made various suggestions for such research. Rather, it is to suggest ways that research on how the phenotype arises from the genotype may be carried forward.

In this article, we first describe the PWS phenotypes before continuing to what is known of the gene(s) whose absence of expression results in PWS and, in particular, the single gene now thought most likely to lie at the heart of the syndrome. We identify what we consider to be particular key questions that need to be addressed and suggest hypotheses as to how the absence of expression of this gene gives rise to the phenotype. We propose that this is either due to a direct effect on brain development, particularly affecting the hypothalamus [2] by impacting on foetal nutrition or by failing to modify the expression levels of a cascade of other (as yet unidentified) genes, which in turn affect brain development. The ultimate effect of the pattern of atypical brain development specific to PWS is to alter the response thresholds for particular networks in the brain thereby impairing the ability to maintain homeostasis, such as ensuring energy balance or responding to environmental change. Finally, based on these hypotheses we suggest new areas for research.

## 2. PWS

PWS arises from the loss of maternally imprinted genes from the paternal chromosome 15 in the region q11-q13. There are two main subtypes, resulting respectively from a deletion in the region q11-q13 (deletion subtype) or from the inheritance of two maternally marked chromosome 15s and no paternally marked chromosome 15 (maternal disomy subtype, (mUPD) or imprinting centre deficit (IC)). The early phenotype is characterised by restricted foetal movement, evidence of foetal growth retardation, severe hypotonia and failure to thrive. The infant with PWS is described as then passing through a period of normal growth and feeding, to a gain in weight not obviously related to an increase in food intake, to an obsessive interest in food in which restriction of food access is necessary to prevent obesity [3]. Later developmental delay, cognitive impairments, short stature due to low growth hormone (GH) levels, and impaired sexual development (undescended testes in males and small labia in females apparent at birth) as well as continued propensity to obesity in the absence of intervention are recognised as part of the PWS phenotype. Ritualistic behaviours, severe emotional outbursts, skin picking, an abnormal sleep pattern, poor temperature regulation and a high pain threshold are also frequently seen. Those with PWS due to the maternal disomy subtype are also more prone to develop a psychotic illness from teenage years on (very high lifetime risk) and more likely to be diagnosed with autism [4,5]. While the diagnostic criteria listed by [6] are extensive, some signs and symptom cluster together and can be explained by a single abnormality. For example, children with PWS who have been on growth hormone since early childhood not only have a normal height they also have normal sized hands and feet and the facial characteristics associated with PWS are much less marked [7]. It is the changing core phenotype, from a foetus and infant that is undernourished, fails to thrive, and is sexually immature to a child who develops hyperphagia, has relative sex and growth hormone deficiencies and developmental delay and intellectual impairments, that ultimately has to be explained by the genetics. We note that the foetus appears to develop normally up to the point where brain development is predominant, in that internal organs are not impaired and it is in the second half of pregnancy that the first symptoms (eg reduced foetal movement) appear.

## 3. PWS Gene

The genes involved in PWS are imprinted, but we do not know for certain how many such genes exist in the PWS region. Much progress on the functions of these genes has been made using mouse models. Mouse chromosome 7 is similar to human chromosome 15, but there is not a perfect match, as shown by the CI5orf2 gene, which has no mouse counterpart [8]. Knock-out mouse models and studies of people with mutations in the MAGEL2 (i.e., Schaaf-Yang syndrome) and/or NECDIN genes suggest that these genes are involved in hypotonia, respiratory problems, sleep abnormalities, adiposity, developmental and cognitive delay, socialisation difficulties, and skin picking [9,10,11,12]. MAGEL2 knock-out mice show the poor suck/failure to thrive characteristic of PWS and the demonstration that in these mice, oxytocin administration in the first five hours after birth restored normal suckling [13] raised hopes of a treatment in humans. However, reviews of trials of oxytocin administration in humans reported mixed results [14,15]. Moreover, two people have been described lacking expression of the genes MKRN3, MAGEL2, and NECDIN but showing only developmental and cognitive delay from the major PWS criteria (see [6] for consensus diagnostic criteria) and a high pain threshold from minor criteria [16,17]. Furthermore, accounts in the literature of people with very small deletions, not involving MKRN3, MAGEL2, or NECDIN, but with most or all PWS characteristics also support the non-involvement of these genes in the PWS core characteristics in humans [18,19,20]. Core characteristics in these cases were: hypotonia, feeding difficulties, hyperphagia/obesity, hypogonadism, intellectual disability, and behaviour problems. The most likely single candidate gene, from studies of people with PWS, is now considered to be SNORD116 but mouse models are inconsistent: [21] support this hypothesis, [22] question it, and others suggest that IPW is also involved [23]. Summarising this evidence, it seems that findings in humans and mice differ, and that, in humans, the gene(s) responsible for the core characteristics of PWS is, or lies in the vicinity of, SNORD116. Further support for this hypothesis comes from the observation that the brain is the only major organ involved in the syndrome. Organs that develop earlier in gestation, such as the heart and lungs, are normal, suggesting that the gene, whose absence of expression results in PWS, is expressed in the brain and not in these other organs.

## 4. The SNORD116 Gene

The gene cluster SNORD116 encoding small nucleolar RNAs is considered to be a cluster of orphan C/D box snoRNAs since it does not target rRNAs or snRNAs. It is expressed prevalently in the brain and lacks any significant complementarity with ribosomal RNA. Due to its affinity with the brain, most studies are performed in mice. Although SNORD116 contains major sequences that are conserved across a number of species, there are some nucleotide differences between human and mouse [24] SNORD116 changes the expression levels of multiple genes [25], which may explain why, in PWS, its absence can affect so many areas. In mouse models selective deletion of SNORD116 from NPY expressing neurons resulted in upregulation of NPY mRNA and low birth weight, increased weight gain in early adulthood, increased energy expenditure, and hyperphagia [26].

## 5. Genotype to Phenotype

As noted above, in PWS threshold levels for several characteristics are changed; this means that the distribution of the particular characteristic in people with PWS is shifted relative to the distribution of that characteristic in the typically developing population towards the more deleterious end. However, significantly, no characteristic is entirely absent or 100% dominant. This might be explained by the relevant gene-controlled threshold levels for these characteristics being changed, exactly what we would expect of a gene that altered expression levels in a number of other genes but did not entirely eliminate expression in these genes. 

One possible version of our single gene hypothesis follows from our earlier observations: SNORD116 changes the expression levels of genes involved in several different regulatory processes (satiation, pain, sleep, temperature) and its absence, as in the case of NPY (related to satiation), causes an increase in the threshold level at which satiety is reached. As [27,28] noted in the case of satiety, the threshold for reaching satiety is increased but satiation is not completely absent in PWS. The persistence of satiation has been demonstrated recently [29]. Thus, SNORD116 appears to be a regulatory gene that regulates other genes by changing their expression levels and its absence therefore changes expression levels from those observed in the non-PWS population in the opposite direction. With this model PWS can therefore be regarded as fundamentally a single gene disorder but it may also be considered to be a polygenic disorder, involving respectively a single gene from the PWS region and multiple genes not yet identified from other areas of the genome. 

Another possible explanation of the genotype to phenotype transition is outlined in [1]. Prader-Willi syndrome arises as a consequence of absent paternal copies of maternally imprinted genes at 15q11-13. Such gender-of-origin imprinted genes are expressed in the brain and also in mammalian placenta where paternally expressed imprinted genes drive foetal nutritional demand. We hypothesise that the PWS phenotype is the result of the genotype impacting two pathways: firstly, directly on brain development and secondly, on placental nutritional pathways that results in its down-regulation and relative foetal starvation. The early PWS phenotype establishes the basis for the later characteristic phenotype. Hyperphagia and other phenotypic characteristics arise as a consequence of impaired hypothalamic development. Hypothalamic feeding pathways become set in a state indicative of starvation, with a high satiety threshold and a dysfunctional neurophysiological state due to incorrect representations of reward needs, based on inputs that indicate a false requirement for food. In this model PWS might be considered to be a single gene disorder in which the phenotype arises as a consequence of abnormal brain development driven by the absence of expression of SNORD 116. A paper by [2] reported on recent neuroimaging findings showing significantly smaller hypothalamic nuclei in people with PWS compared to an aged-matched typically developing control group and also an obese group. The authors argue that the hypothalamus fails to develop in PWS and this would explain the varied phenotypic characteristics of hypothalamic origin.

## 6. Research Directions

The above hypotheses give rise to possible fruitful research directions: firstly, investigation of genes whose expression levels are changed by SNORD116 including investigation of PWS placental genes, especially those whose expression levels are changed by SNORD116. Secondly, using mouse models, the investigation of placental foetal nutritional pathways to see if these are down-regulated, and thirdly; from a phenotypic perspective, to take a symptom such as high pain threshold, whose extreme would be a complete inability to feel pain, as in Congenital Insensitivity to Pain (CIPA), for example, whose cause is the inheritance of two copies of the NTRK1 gene [30]. Then ask the questions: Would over-expression of a single copy lead to a higher pain threshold and does SNORD116 alter the expression level of this gene? FOURTHLY, from the observation that the motor cortex and hypothalamus are involved in PWS symptoms, are there any genes specific to these areas? Are the expression levels of any such genes changed by the absence of SNORD116 expression?

The genotype to phenotype pathways in PWS remain an enigma. Fundamental to understanding the syndrome and to the development of new treatments is understanding how the genotype leads to the initial failure to thrive phenotype and then how this then results in the characteristic PWS phenotype. We challenge the accepted wisdom on this proposing two contrasting perspectives. We hope that this article will help to stimulate research.

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
