# Peer review of "Next Steps in Prader-Willi Syndrome Research: On the Relationship between Genotype and Phenotype"

_ijms, 2022, doi:10.3390/ijms232012089_

Round 1

Reviewer 1 Report

A nice and concisely written hypothesis. I have no comments. The current manuscript addresses the pathogenesis of PWS: how do the genetic changes give rise to the phenotype. This topic is interesting for the readership and is a bit provoking, while the authors pose their opinion. The paper is well written and scientifically sound. While it is a paper stating a hypothesis the paper is correct and reflects the opinion of the authors. Therefore a have no comments or suggestions for improving the paper.

Author Response

We thank the reviewer for the very positive review.

Reviewer 2 Report

The manuscript is well thought out and the hypotheses considered are very interesting. I strongly support the Authors to report their suggestions. I have just a few suggestions for improving the paper.

1) Title: The title should specify that the topic of the paper is the next steps in PWS research on the relationship between genotype and phenotype. There are indeed many other areas where further research is needed.

2) Page 2 lines 6-9: the description of the “later” PWS phenotype should include  the presence of early severe obesity (if uncontrolled) and its comorbidities.

3)- Page 2, line 40: A recent review on oxytocin-based therapies for treatment of PWS should be added in the references: Althammer F, Muscatelli F, Grinevich V, Schaaf CP. Oxytocin-based therapies for treatment of Prader-Willi and Schaaf-Yang syndromes: evidence, disappointments, and future research strategies. Transl Psychiatry. 2022 Aug 8;12(1):318.

4)- Page 3 line 32: the persistence of satiation in PWS has been demonsrated by Rigamonti et al. (Rigamonti AE, Bini S, Grugni G, Agosti F, De Col A, Mallone M, Cella SG, Sartorio A. Unexpectedly increased anorexigenic postprandial responses of PYY and GLP-1 to fast ice cream consumption in adult patients with Prader-Willi syndrome. Clin Endocrinol (Oxf). 2014 Oct;81(4):542-50.

5)- Page 4 line 3:  “Brown et al (in press)” should be replaced by “Brown et al. 2022”.

6)- Page 4 line 18: Thirdly is a repetition; please, replace with Fourthly.

7)- References: Please standardize all references according to the indications of the Journal.

Author Response

We thank the reviewer for the positive review and for the helpful comments enabling us to improve the document.

We have complied with all suggestions: 

The title has been amended as suggested

p2 line6 text added in upper case

p2 line 40 reference added

 p3 line 32 text in upper case and reference added

p4 line 3 text replaced by reference number

p4 line 18 thirdly replaced by Fourthly

References in text and in reference section altered in line with journal recommendations.